# ClimateBERT-NetZero: Detecting and Assessing Net Zero and Reduction Targets

**Tobias Schimanski**[1] **Julia Bingler**[2,3] **Camilla Hyslop**[2,4] **Mathias Kraus**[5] **Markus Leippold**[2,6]

[1]University of Zurich [2]University of Oxford [3]Council on Economic Policies
[4]Net Zero Tracker [5]FAU Erlangen-Nürnberg [6]Swiss Finance Institute (SFI)
`tobias.schimanski@bf.uzh.ch`

## Abstract

Public and private actors struggle to assess the vast amounts of information about sustainability commitments made by various institutions. To address this problem, we create a novel tool for automatically detecting corporate, national, and regional net zero and reduction targets in three steps. First, we introduce an expert-annotated data set with 3.5K text samples. Second, we train and release ClimateBERT-NetZero, a natural language classifier to detect whether a text contains a net zero or reduction target. Third, we showcase its analysis potential with two use cases: We first demonstrate how ClimateBERT-NetZero can be combined with conventional question-answering (Q&A) models to analyze the ambitions displayed in net zero and reduction targets. Furthermore, we employ the ClimateBERT-NetZero model on quarterly earning call transcripts and outline how communication patterns evolve over time. Our experiments demonstrate promising pathways for extracting and analyzing net zero and emission reduction targets at scale.

## 1 Introduction

**Motivation.** "Zero tolerance for net zero greenwashing" is the central aim of UN General Secretary Antonio Guterres at the GOP27 (UN, 2022). To limit global warming to 1.5°C, massive efforts in emission reduction are necessary (Klaassen and Steffen, 2023). Consequently, an increasing amount of net zero and emission reduction targets are introduced, communicated, or updated by private and public institutions (Höhne et al., 2021). With the help of Natural Language Processing (NLP) methods, it is possible to automatically assess large chunks of textual data and gain structured insights about climate-related information (e.g., Webersinke et al., 2022; Bingler et al., 2022a; Stammbach et al., 2022). However, none of the previous works have studied the arguably most relevant and best tractable objective for institutions

– the extraction of information about net zero and reduction targets.

**Contribution.** As a remedy, this study delivers a threefold contribution at the intersection of climate change and NLP by creating a tool that automatically extracts and assesses net zero and reduction target information from various sources. Our first contribution is introducing an expert-annotated data set with 3.5K text samples, which builds on the Net Zero Tracker project (Lang et al., 2023). Second, we develop and publish ClimateBERT-NetZero (based on Climate-BERT (Webersinke et al., 2022)), a comparatively lightweight and, therefore, less energy-intensive NLP model that can effectively classify net zero and reduction targets in text samples.[1] Third, we provide two real-world use cases: We demonstrate how to extend the model capabilities and analyze the ambitions of the net zero and reduction targets. Finally, we utilize ClimateBERT-NetZero to analyze the appearance of net zero and reduction claims in companies' earning calls transcripts from 2003 to 2022.

**Results.** In this context, we show that ClimateBERT-NetZero achieves better predictive performance than larger models. Moreover, we demonstrate that we can effectively detect the target year for net zero claims; as well as the target year, baseline year, and reduction targets in percent for general emission reduction claims. Finally, we provide evidence for the rise in target communications by analyzing communicated net zero and reduction targets in companies' earning calls.

**Implications.** The implications of our study are significant for both practice and research. Analyzing net zero and reduction targets represents a crucial step in understanding and acting on transition plans. Since the tool can operate on a large scale, we lower the barriers to data accessibility.

---

[1]Model and data are available on `https://huggingface.co/climatebert/netzero-reduction`.

Acknowledging that further research is needed to hold companies responsible for their commitments and analyze potential greenwashing patterns, our approach presents an important first step towards "zero tolerance for net zero greenwashing".

## 2 Background

**Large Language Models.** In previous years, Large Language Models (LLMs) continuously gained importance. LLMs have proved to possess human-like abilities in a variety of domains (Radford et al., 2019; Brown et al., 2020; Ouyang et al., 2022). These models are primarily based on subtypes of the transformer architecture and are trained on massive amounts of data to learn patterns within the training data (Vaswani et al., 2017). As a result of the training process, the models can solve a multitude of downstream tasks. One particular downstream task represents text classification. For these tasks, commonly used models are based on the Bidirectional Encoder Representations from Transformers (BERT) architecture (Devlin et al., 2019).

**NLP in the Climate Change Domain.** Our work contributes to an ongoing effort to employ LLMs to support climate action by developing targeted datasets and models. Previously released work includes the verification of environmental claims (Stammbach et al., 2022), climate change topic detection (Varini et al., 2021), and the analysis of climate risks in regulatory disclosures (Kölbel et al., 2022). A particular way to improve the performance of models on downstream tasks is the creation of domain-specific models, such as ClimateBERT. With further pre-training on climate-related text samples, this model outperforms its peers trained on general domain language (Webersinke et al., 2022). ClimateBERT has already been applied in recent studies that analyze corporate disclosures to investigate patterns around companies' cheap talk (Bingler et al., 2022a,b).

Apart from model-centric considerations and innovations, previous research and public attention have also shifted to the climate impact of training and deploying LLMs (Stokel-Walker and Noorden, 2023; Hao, 2019). With the ongoing development of new models, the models' number of parameters steadily rises. Consequently, this increases electricity requirements and, therefore, the environmental footprint of training and using the models. For instance, while ClimateBERT possesses 82 million parameters (Webersinke et al., 2022), ChatGPT has up to 175 billion parameters (OpenAI, 2023). Hence, climate-aware decisions for model training and deployment must be carefully considered.

**Availability of Climate Change Data.** In light of continuously increasing societal and regulatory pressure for firms to disclose their climate change activities, data availability will drastically increase in the upcoming years. For instance, the EU requires companies to rigorously disclose risks and opportunities arising from environmental and social issues from 2025 on (EU, 2023). An important role is attributed to credible and feasible transition plans aligning with the ambition to restrict global warming to 1.5 degrees (Höhne et al., 2021; Fankhauser et al., 2022). Thus, assessing firms' reduction or net zero commitments plays an essential role. To this end, the present study introduces a data set and methods to automatically detect and assess reduction and net zero statements by private and public institutions.

## 3 Data

The raw data for this study originates from a collaboration with the Net Zero Tracker project (Lang et al., 2023). The Net Zero Tracker assesses targets for reduction and net zero emissions or similar aims (e.g., zero carbon, climate neutral, or net negative). Reduction targets are claims that refer to an absolute or relative reduction of emissions, often accompanied by a baseline year to which the reduction target is compared. Net zero targets represent a special case of reduction targets where an institution states to bring its emissions balance down to no additional net emissions by a certain year. The Net Zero Tracker assesses claims of over 1,500 countries, regions, cities, or companies.[2]

For this project, we created an expert-annotated 3.5K data set. The data set differentiates between three classes: net zero, reduction, and no targets. For the reduction and net zero targets, we further process the claims gathered by the Net Zero Tracker. If necessary, we clean and hand-check the text samples and correct the pre-assigned labels by the Net Zero Tracker. Moreover, we enhance the data with a variety of non-target text samples. All of these samples originate from annotated and reviewed datasets of previous projects in the climate domain (Stammbach et al., 2022; Webersinke et al.,

---

[2]Collectively, the dataset contains 273 claims by cities, 1396 claims by companies, 205 claims by countries, and 159 claims by regions.

| Net Zero | Reduction | No Target |
|----------|-----------|-----------|
| 990 | 1005 | 1522 |

Table 1: Distribution of the labels in the text samples

| Model | Model size | Acc (std.) |
|-------|-----------|------------|
| ClimateBERT | 82 M | 0.966 (.004) |
| DistilRoBERTa | 82 M | 0.959 (.007) |
| RoBERTa-base | 125 M | 0.963 (.006) |
| GPT-3.5-turbo | 175 B | 0.938 |

Table 2: Accuracy and model size of tested models

2022) and are again hand-checked by the authors (for more details, see Appendix A). Ultimately, we construct a dataset with 3.5K text samples (see Table 1).

# 4 ClimateBERT-NetZero

We use the data set to train and test the capabilities of several classifier models. We particularly want to focus on training a model that is both effective in performance and efficient in the use of resources. Thus, as a base model for fine-tuning, we utilize ClimateBERT (Webersinke et al., 2022). This model is based on DistilRoBERTa which is a smaller and thus less resource-demanding version of RoBERTa (Sanh et al., 2020). ClimateBERT is further pre-trained on a large corpus of climate-related language and thus outperforms conventional models on a variety of downstream tasks (Webersinke et al., 2022). We further fine-tune ClimateBERT to classify whether a text contains a net zero target, a reduction target, or none of them. We christen this model ClimateBERT-NetZero. To set the results into context, we rerun the fine-tuning with its base architecture DistilRoBERTa as well as with the larger model RoBERTa (Liu et al., 2019). Furthermore, we also test how generative, much more resource-intensive, but also more general models perform in comparison to ClimateBERT-NetZero. To create this comparison, we prompt GPT-3.5-turbo to classify the texts (see Appendix B for the detailed prompt).

To evaluate the performance of the BERT-family models, we run a five-fold cross-validation and consult the performance measures accuracy, F1-score, precision, and recall (see Appendix C for all explored variations of hyperparameters and Appendix D for full results and description of the validation set). As Table 2 shows, ClimateBERT-NetZero slightly outperforms all other BERT-family models. With a consistent accuracy of over 96%, ClimateBERT-NetZero is able to distinguish between the different types of targets effectively. Furthermore, we evaluate the performance of GPT-3.5-turbo by prompting it to classify all texts, transform the output to numeric labels, and calculate the performance metrics. As Table 2 indicates,

ClimateBERT-NetZero also outperforms GPT-3.5-turbo on this task.[3]

To further evaluate the performance of the ClimateBERT-NetZero in a real-world setting, we conduct a hand-check on the output of the model on 300 sentences sampled from sustainability reports. We find that the model classifies 98% of the sentences right. This solidifies the robustness of the approach (see Appendix E for more details).

Since net zero targets are a special case of reduction targets, we also create a model called ClimateBERT-Reduction. This model simply differentiates between general reduction targets (including net zero targets) and no targets. We repeat the previous setup to train and test the model. ClimateBERT-Reduction consistently reaches performance measures of ca. 98% (for more details, see Appendix F).

# 5 Use Cases

To showcase further capabilities of ClimateBERT-NetZero, we develop two potential use cases. First, we showcase how the model can be used to perform a deeper analysis of net zero and reduction targets by analyzing the underlying ambitions of the statements. Second, we demonstrate how ClimateBERT-NetZero can detect patterns in company communication by analyzing earning call transcripts.

**Measure Ambitions with Q&A Models.** In this use case, we study methods to assess the ambitions of institutions further. For net zero claims, the primary measure of ambition is given by the year the institution wants to achieve net zero. For reduction claims, we use the target year of the reduction, the percentage of how much the institution wants to decrease its emissions, and the baseline year against which the reduction is measured.

For evaluating the ambitions in the net zero and reduction claims, we employ the Question Answering (Q&A) model Roberta-base-squad2 (Rajpurkar et al., 2016). This model delivers concise answers

---

[3]Model and data are available on `https://huggingface.co/climatebert/netzero-reduction`.

to questions posed to a text sample. Therefore, we develop questions to extract the relevant information for each dimension of ambition in the net zero and reduction claims (see Appendix G for detailed questions).

To evaluate this approach, we again use the data compiled by the Net Zero Tracker project (Lang et al., 2023). The data set contains human-extracted data for each of the researched dimensions of ambitions. To test the ambitions, we preprocess and hand-check 750 text samples.

Finally, we run the Q&A model on the text samples. The model typically outputs a single number. We assess the accuracy of the model by checking how many answers contain the correct number. Table 3 shows the results of each dimension in three steps. First, we evaluate the question on the raw data set. For approximately 95% of the net zero targets, the model can detect by which year net zero is aimed to be achieved. In the second step, we reduce the data set so that a target text is only included if the text actually contains the information we search for. This gives the upper bound accuracy of the model. For instance, this reduces the data set for the baseline year detection of reduction targets to 84% while increasing the accuracy from 67% to 80%. This underlines the potential for improving Q&A results with more fine-grained data cleaning. In the third step, we again use the raw data and improve the performance by using the confidence of the Q&A model. Optimizing the confidence represents a trade-off between boosting the accuracy and preserving an amount of data where these confident decisions can be taken (see Appendix H for more details). Table 3 reports the results using a confidence of 0.3. Increasing the model confidence consistently improves the accuracy.

**Detection Net Zero and Reduction Targets in Earning Call Transcripts.** In this use case, we employ ClimateBERT-NetZero on transcripts of quarterly earning calls from publicly-listed US companies from 2003–2022 to demonstrate its large-scale analysis potential. We create a yearly index by building an average of all quarters of a year. As Figure 1 indicates, the net zero target discussion has experienced a sharp increase since 2019.[4] While the same observation is true for reduction targets,

---

[4]This observation largely coincides with the general large increase in sustainable investments in 2019–2022 (see, for instance https://www.morningstar.com/sustainable-investing/2022-us-sustainable-funds-landscape-5-charts)

| Task | Data | Accuracy (% of data) |
|------|------|----------------------|
| **net zero target year** | raw/optimal | 0.95 (1.0) |
| | confidence-tuned | 0.97 (0.5) |
| **reduction target year** | raw | 0.84 (1.0) |
| | optimal | 0.90 (0.97) |
| | confidence-tuned | 0.90 (0.8) |
| **reduction base year** | raw | 0.68 (1.0) |
| | optimal | 0.80 (0.84) |
| | confidence-tuned | 0.82 (0.62) |
| **reduction percentage** | raw | 0.88 (1.0) |
| | optimal | 0.91 (0.96) |
| | confidence-tuned | 0.93 (0.35) |

Table 3: Accuracy of the Q&A approach in assessing the quantitative properties of the net zero and reduction targets

they seem to be discussed throughout the whole period. These results underline both the rising importance of the topic in recent times as well as the value of differentiating between net zero and reduction targets (for further details, see Appendix I).

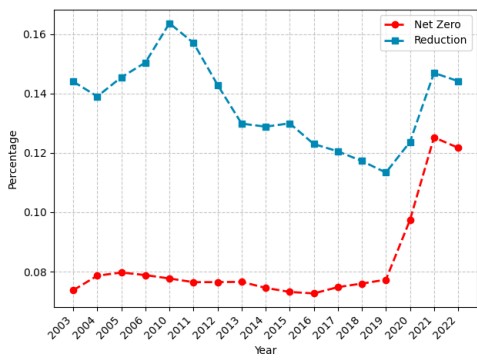

Figure 1: Communication of net zero and reduction target sentences as a percentage of all sentences over time

## 6   Conclusion

In conclusion, this paper demonstrates the development and exemplary employment of ClimateBERT-NetZero, a model that automatically detects net zero and reduction targets in textual data. We show that the model can effectively classify texts and even outperforms larger, more energy-consuming architectures. We further demonstrate a more fine-grained analysis method by assessing the ambitions of the targets as well as demonstrating the large-scale analysis potentials by classifying earning call

transcripts. By releasing the dataset and model, we deliver an important contribution to the intersection of climate change and NLP research. Identifying and analyzing reduction and net zero targets represent an important first step to controlling set commitments and further comparing them with other real-world indicators of corporate, national, or regional actors. Especially in the realm of greenwashing, the abundance or failure to achieve set targets while benefitting from a differently communicated public image are key factors to investigate.

# 7 Limitations

Like all studies, ours also has limitations. First, the dataset size is limited to 3.5K samples. While this could entail lower generalizability, we show that the model performance is already at a very high level. The data size limitation also applies to the first use case, where further data could increase the validity. Second, this study predominately focuses on BERT-family models. In a strive to employ accessible models of smaller size, we do not employ larger, more recent architectures such as LLaMa (Touvron et al., 2023). While the impact on the classification task might be limited, as shown with GPT-3.5-turbo, especially for detecting ambitions, instruction-finetuned models could improve the results.

## Acknowledgements

This paper has received funding from the Swiss National Science Foundation (SNSF) under the project 'How sustainable is sustainable finance? Impact evaluation and automated greenwashing detection' (Grant Agreement No. 100018_207800).

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

# Appendix

## A    Details on Training Data Creation

The training data consists of three parts: reduction targets, net-zero targets, and non-target texts.

**Reduction and Net-Zero Targets.** For reduction and net-zero targets, we collaborate with the Net Zero Tracker project (Lang et al., 2023). The project tracks the commitments of over 1,500 companies, countries, regions, and cities.[5] We use their raw data and process it in three steps.

First, we transform the raw data in several ways. The Net Zero Tracker follows the general classification of reduction and net zero targets but uses a more fine-granular classification. For instance, net-zero targets could be zero carbon, climate neutral, or net negative. We aggregate all sub-labels into their two main categories. Furthermore, we preprocess the corresponding texts. We remove sentences shorter than five words, URLs, and other special characters. However, these cleaning steps only result in marginal changes in the data characteristics, reducing the overall dataset by 97 samples (see Table A.1).

Second, we hand-check the labels of the remaining text samples. Since the texts are already assigned with labels gathered by expert annotators of the Net Zero Tracker project.[6] The aim of this step is to increase validity by hand-checking the labels. This step creates an inter-labeler agreement (see Table A.2). The results show a high inter-labeler agreement with a Cohen's Kappa of 93%. We correct labels if necessary or remove text samples that don't correspond to their labels or are meaningless. Each text sample is hand-checked by at least one author. If the label is found to be different from the expert annotators of the Net Zero Tracker, the decisions are marked and decided upon by the team. Thus, we create two labels: First, the labels assigned by our data collaborators of the Net Zero Tracker, and second, the labels assigned by the author team. For a unified decision-making process, the following labeling guidelines are used:

*Reduction targets are claims that refer to an absolute or relative reduction of emissions, often accompanied by a baseline year to which the reduction target is compared.*
*Net zero targets represent a special case of reduction targets where an institution states to bring its emissions balance down to no additional net emissions by a certain year.*
*If both targets appear in the text, the main focus of*

---

[5]Further information can be found on the Net Zero Tracker website https://zerotracker.net/.

[6]The Net Zero Tracker is a combined initiative of the University of Oxford Net Zero, the New Climate Institute, Data Driven Envirolab, and the Energy & Climate Intelligence Unit. Thus, the annotators we collaborate with in this project are domain experts.

*the text is decisive. For instance, most reduction targets serve as intermediary steps for the final goal of net zero. Thus, the focus lies on net zero.*

The following examples provide an impression of a reduction and net zero target:

```
Reduction target: 'We will reduce our absolute
    Scope 1 and 2 GHG emissions from non-
    locomotive operations (including emissions
    associated with our buildings and
    facilities) by 27.5% by 2030.'
Net zero target: 'We are committed to net-zero
    energy across our businesses and achieved
    100% carbon neutrality for our operations
    by purchasing carbon offsets with Renewable
    Energy Certificates (REC).'
```

Third, we use a human-in-the-loop validation process to enhance the validity of the data further. We use five-fold-crossvalidation with the baseline ClimateBERT model on the entirety of the dataset and save the missclassifications of each fold. This gives us the opportunity to train the model with the data and, at the same time, store the wrongly identified samples within the training process. We then involve a human annotator to check the mis-classifications and readjust labels if necessary. This way, we use the machine to identify edge cases and the human to interfere if needed. We repeat this human-in-the-loop validation three times.

**Non-Target Samples.** For non-target text samples, we use publicly available datasets in the climate domain (Stammbach et al., 2022; Webersinke et al., 2022). We aim to increase the generalizability of the models by creating a well-balanced dataset. Including climate-related samples that are not targets is crucial for the models to learn the specific differentiation between *targets* and *general* sentences in the climate domain. Furthermore, the datasets enable us also to include generic, non-climate-related sentences. We follow the same procedure described above to improve the validity of the dataset.

Ultimately, we create a dataset with 3.5K text samples. Table A.1 gives an overview of the descriptive statistics of the text length.

## B  GPT-3.5-turbo Prompt

To create a comparison between BERT-family and larger instruction-finetuned models, we also use GPT-3.5-turbo to classify all texts. We use the following prompt for zero-shot classifying the dataset to contain a reduction, net zero, or no target at all:

```
"""Your task is to classify a provided text whether
    it contains claims about Reduction or Net Zero
    targets or none of them.

Reduction targets are claims that refer to an
    absolute or relative reduction of emissions,
    often accompanied by a baseline year to which
    the reduction target is compared.
Net zero targets represent a special case of
    reduction targets where an institution states
    to bring its emissions balance down to no
    additional net emissions by a certain year.
If both targets appear in the text, the main focus
    of the text is decisive. For instance, most
    reduction targets serve as intermediary steps
    for the final goal of net zero. Thus, the focus
    lies on net zero.

As an answer to the provided text, please only
    respond with 'Reduction' for reduction targets,
    'Net Zero' for Net Zero targets or 'None' if
    no category applies.

Provided text: ^^^{text}^^^ """
```

## C  Hyperparamters for Fine-Tuning

Table C.3 displays the base case of the configured hyperparameters for fine-tuning ClimateBERT-NetZero, DistilRoBERTa and RoBERTa-base. The performance metrics are created using a five-fold crossvalidation.

Furthermore, Table I.9 systematically explores the variation of the hyperparameters, such as the learning rate, batch size, and epochs. The results show that ClimateBERT consistently outperforms its peers. However, it also shows that the choice of the hyperparameters only marginally affects the final outcome accuracy.

## D  Full Performance Metrics of All Models

For the full performance metrics of all models with the base hyperparameter setting, see Table I.8. ClimateBERT consistently outperforms all other models for the performance metrics accuracy, F1-score, precision, and recall.

Furthermore, the high F1, precision, and recall values suggest that all three labels are consistently predicted well with the models. Figure I.1 solidifies this notion by displaying the confusion Matrix for the cross-validation results of the ClimateBERT model, taking the predictions of each validation set at the end of a fold. All three labels are consistently predicted on a very satisfactory level. The most confusion is visible between a reduction and net zero targets, which are naturally semantically close. In particular, outstanding performance is achieved on the Non-Target label, which is very important

| Processing | count | mean | std | min | max | 25% | 75% |
|---|---|---|---|---|---|---|---|
| **before** | 3614 | 38.5 | 59.0 | 1 | 1057 | 16 | 40 |
| **after** | 3517 | 39.5 | 59.5 | 5 | 1057 | 17 | 40 |

Table A.1: Characteristics of the label data text length before and after the processing steps

| Labler Agreement | Cohen's Kappa |
|---|---|
| 0.952 | 0.931 |

Table A.2: Statistics on labeler agreement

| Hyperparameter | Value |
|---|---|
| Epochs | 10 |
| Batch Size | 32 |
| Gradient Accumulation | 2 |
| Warmup ratio | 0.1 |
| Learning rate | 5e-5 |
| Patience | 5 |

Table C.3: Hyperparamters for training the models with five-fold cross-validation

for real-world applications where this label is likely the overwhelming majority class.

For an overview of the validation data used in the five-fold cross-validation, see Table D.4. Since we use a stratified split, the distribution in each fold is nearly equal.

| Fold | Non-Target | Reduction | Net-Zero |
|---|---|---|---|
| 0 | 305 | 201 | 198 |
| 1 | 305 | 201 | 198 |
| 2 | 304 | 201 | 198 |
| 3 | 304 | 201 | 198 |
| 4 | 304 | 201 | 198 |

Table D.4: Label distribution in the validation sets of the five-fold cross-validation

## E   Hand-Evaluation of Sustainability Reports

In this test, we hand-evaluate 300 sentences of 2022 sustainability reports. We take the sustainability reports of Pfizer, JP Morgan, and Apple (to have a variety of data sources), classify the entirety with ClimateBERT-climate and ClimateBERT-NetZero, and sample 300 sentences for hand-checking. In this test set, we include all sentences labeled as first climate and second net zero or reduction target (63) and fill up the rest of the test set with random

sentences (237). We read every sentence and assign a label to it. The evaluation is largely in line with the overall evaluation metrics. The F1 scores are 98% for net zero, 90% for reduction, and 99% for non-target. The comparatively lower value for reduction results from some confusion between the reduction of waste or water and the reduction of emissions, i.e., a confusion between reduction and no target. Further details can be seen in the corresponding confusion matrix in Figure I.2.

## F   Performance Metrics for ClimateBERT-Reduction

Table F.5 displays the evaluation metrics for the ClimateBERT-Reduction model that defines net zero and reduction targets as a combined class. With consistent measures over 98%, the model can successfully differentiate between the classes.

| Metric | Score |
|---|---|
| # Parameters | 82 million |
| Accuracy (std.) | 0.985 (0.005) |
| F1-Score (std.) | 0.987 (0.005) |
| Precision (std.) | 0.990 (0.004) |
| Recall (std.) | 0.988 (0.005) |

Table F.5: Performance metrics of ClimateBERT-Reduction

## G   Questions for Evaluating the Ambition of Targets

These questions are posed to the text samples to evaluate the ambition of the net zero and reduction targets.

- For detecting the target year of net zero targets:

```
'When does the organization want to
     achieve net zero?'
```

- For reduction targets:

  – For detecting the target year:

```
'By which year does the organization
     want to reduce its emissions?'
```

– For detecting the baseline year:

```
'What is the baseline year or level
    for the target to which the
    reduction target is compared to
    ?'
```

– For detecting the relative or absolute reduction target:

```
'What is the reduction target of the
    organization in % or absolute
    terms?'
```

These questions are posed to the Q&A model Roberta-base-squad2 (Rajpurkar et al., 2016).

## H  Model Accuracy and Dataset Size as a Function of Model Confidence

The graphs in H.4 display the model accuracy and the corresponding remaining fraction of the dataset for a given confidence level of a model. The model accuracy predominately increases when the model confidence increases. At the same time, the size of the dataset is reduced because there are fewer samples for which the model is confident. For instance, for a confidence of 0.3, the accuracy of detecting the baseline year of the reduction target rises from 67% to 82% while still analyzing 62% of the data set.

## I  Details on the Earning Call Transcript Evaluation

The base data for the evaluation are earning conference call (ECC) transcripts originating from Refinitiv. We use all available ECC data for publicly listed firms in the US from 2003-2022. The raw documents are split into single sentences. We assess a total of 17825 earning call events. The summary statistics of the number of sentences per event can be seen in Table I.6. After creating the sentence-level data, we use a two-step process to identify net zero and reduction targets. We first classify whether the sentences are climate-related or not with the ClimateBERT climate detector model.[7] Then, we use ClimateBERT-NetZero to classify all climate-related sentences as either a net zero, reduction, or no target. Especially for the reduction class,

it seems crucial to first identify climate-related sentences. Otherwise, the chance of false positives might be high. ClimateBERT-NetZero might pick up general reduction targets that are semantically similar to non-climate reduction targets. Finally, we calculate the mean for every ECC and aggregate it per year.

The results are displayed in Figure 1. As the Figure indicates, the general discussion of reduction targets occurs throughout the sample period. It even reached its peak around 2010 and has experienced a steady decrease since then. Net zero targets are rarely discussed in the earlier sample periods. However, in 2019, both the reduction and net zero target discussion increased drastically. This likely originates from the general mainstreaming of climate issues in recent years. However, this analysis represents a preliminary step and should inspire other researchers to use the models and find patterns. It demonstrates the large-scale analysis capabilities of the ClimateBERT-NetZero.

---

[7]See        https://huggingface.co/climatebert/distilroberta-base-climate-detector for the model.

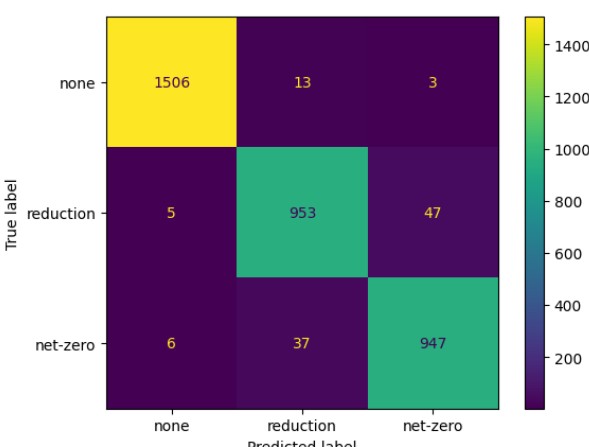

Figure I.1: Confusion Matrix for the cross-validation of the ClimateBERT-NetZero model extracting the results on the validation set for each fold

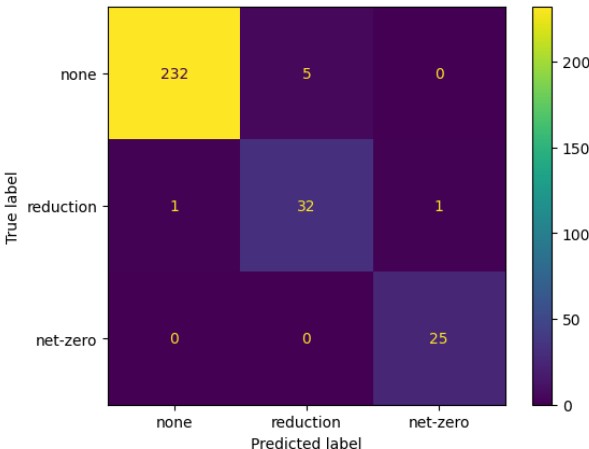

Figure I.2: Confusion Matrix for the hand-checking of the classification results of ClimateBERT-NetZero on 2022 sustainability reports from Pfizer, JP Morgan and Apple

| Sentences per Event | # events | mean | std | min | max | 25% | 50% | 75% |
|---|---|---|---|---|---|---|---|---|
| | 17825 | 329 | 149 | 5 | 1799 | 217 | 323 | 428 |

Table I.6: Summary statistics of the number of sentences in an earning call event

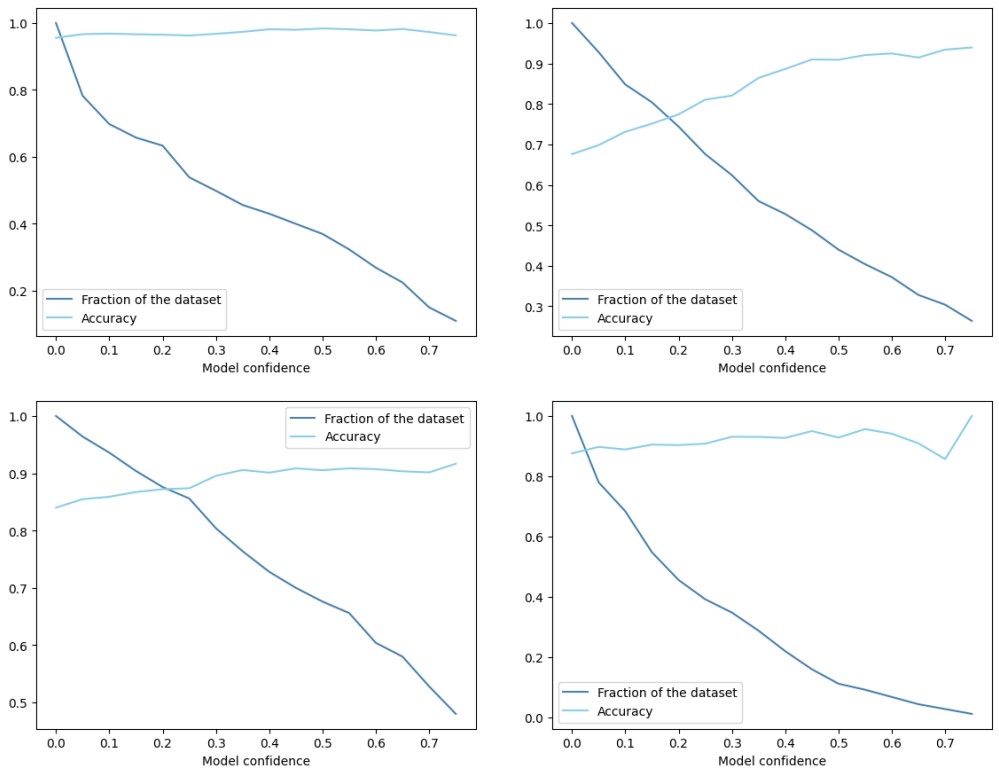

Table I.7: Top left: Accuracy of the net zero target year detection as a function of the model confidence, Top right: Accuracy of the baseline year detection as a function of the model confidence, Bottom left: Accuracy of the reduction target year detection as a function of the model confidence, Bottom right: Accuracy of the relative or absolute target detection as a function of the model confidence

| Model | Model size | Accuracy (std.) | F1 -Score (std.) | Precision (std.) | Recall (std.) |
|---|---|---|---|---|---|
| ClimateBERT | 82 million | 0.966 (.004) | 0.962 (.004) | 0.962 (.004) | 0.963 (.004) |
| DistilRoBERTa | 82 million | 0.949 (.007) | 0.944 (.005) | 0.955 (.007) | 0.946 (.01) |
| RoBERTa-base | 125 million | 0.963 (.006) | 0.958 (.006) | 0.959 (.007) | 0.958 (.006) |
| GPT-3.5-turbo | 175 billion | 0.938 | 0.920 | 0.932 | 0.901 |

Table I.8: Performance metrics of tested models

| Hyperparamters | | | Model Accuracy (std.) | | |
|---|---|---|---|---|---|
| Learning rate | Epochs | Batch Size | ClimateBERT | DistilRoBERTa | RoBERTa-base |
| 3e-05 | 5 | 16 | **0.965 (.005)** | 0.963 (.005) | **0.965 (.005)** |
| 3e-05 | 5 | 32 | **0.971 (.005)** | 0.963 (.005) | 0.965 (.005) |
| 3e-05 | 10 | 16 | **0.963 (.005)** | 0.958 (.005) | 0.956 (.003) |
| 3e-05 | 10 | 32 | **0.968 (.002)** | 0.961 (.003) | 0.964 (.003) |
| 5e-05 | 5 | 16 | **0.965 (.003)** | 0.963 (.007) | 0.962 (.004) |
| 5e-05 | 5 | 32 | **0.967 (.006)** | 0.963 (.005) | 0.964 (.005) |
| 5e-05 | 10 | 16 | **0.963 (.002)** | 0.957 (.005) | 0.961 (.006) |
| 5e-05 | 10 | 32 | **0.966 (.004)** | 0.959 (.007) | 0.963 (.006) |
| 7e-05 | 5 | 16 | **0.966 (.003)** | 0.962 (.006) | 0.962 (.007) |
| 7e-05 | 5 | 32 | **0.966 (.004)** | 0.960 (.005) | 0.966 (.007) |
| 7e-05 | 10 | 16 | 0.961 (.005) | **0.962 (.006)** | 0.959 (.006) |
| 7e-05 | 10 | 32 | **0.965 (.001)** | 0.963 (.002) | 0.964 (.005) |

Table I.9: Performance metrics of the models for a grid of varying hyperparameters learning rate, batch size and epochs