# OpenReview forum: "ClimateBERT-NetZero: Detecting and Assessing Net Zero and Reduction Targets"
_EMNLP/2023/Conference — EMNLP 2023 Main_

### Official Review · Reviewer_vRSM · 2023-07-25

**Soundness:** 3

**Excitement:**

3: Ambivalent: It has merits (e.g., it reports state-of-the-art results, the idea is nice), but there are key weaknesses (e.g., it describes incremental work), and it can significantly benefit from another round of revision. However, I won't object to accepting it if my co-reviewers champion it.

**Missing References:**

1. RoBERTa model is used but not cited: Yinhan Liu, Myle Ott, Naman Goyal, Jingfei Du, Mandar Joshi, Danqi Chen, Omer Levy, Mike Lewis, Luke Zettlemoyer, and Veselin Stoyanov. 2019. Roberta: A robustly optimized bert pretraining approach. arXiv preprint arXiv:1907.11692.

    2. Anna Rogers article: Anna Rogers, Niranjan Balasubramanian, Leon Derczynski, Jesse Dodge, Alexander Koller, Sasha Luccioni, Maarten Sap, Roy Schwartz, Noah A. Smith, and Emma Strubell. 2023. Closed ai models make bad baselines.

**Paper Topic And Main Contributions:**

The research article focuses on enhancing the existing Net Zero Tracker's capability to algorithmically assess the sustainability targets (reduction and net zero goals) of corporations and nations. To accomplish this, the authors present a labeled dataset, fine-tune a language model known as ClimateBERT-NetZero, and aim to make this trained model publicly available.

The research further introduces two unique use cases for the fine-tuned model (ClimateBERT-NetZero). First, it employs the extracted climate claims to ask supplementary questions and garner information such as the target year, base year, and percentage reduction. Second, it utilizes the trained model to extract climate goals from earnings call transcripts, illustrating that the percentage of companies communicating climate goals saw a significant rise in 2020 and 2021.

**Questions For The Authors:**

Q1: What is the distribution of data between companies, countries, regions, and cities categories? Table 1 doesn’t provide detail on this.
Q2: Appendix lines 451-455: How does performed preprocessing change the descriptive statistics of the data? Does it induce any bias?
Q3: Appendix lines 456-463: If the data was already labeled by Net Zero Tracker then how can paper stat contribution as introducing new data? It is more appropriate to say refine the data.
Q4: Appendix lines 463-464: What do you mean by “at least one author”? What was the average number of annotators used for each data point? How it was decided how many annotators to use for a particular data point.
Q5: The paper claims that data was annotated by experts. What was the qualification of the annotators?
Q6: What was an inter-annotator agreement?
Q7: Appendix lines 482-491: The entire process of human-in-the-loop is not clear. Which model was used to identify misclassification? Filtering sentences for readjusting labels based on the model’s output might induce some kind of bias. If the model is wrong on certain data points which can be also wrongly labeled in the first place as authors are refining them, it contaminates the labeled dataset.
Q8: What is the accuracy of the model in detecting Net Zero and Reduction Targets in Earnings Call Transcripts?
Q9: In Figure 1, does y-axis values are multiplied by 100 to show the % value or it is just a fraction?
Q10: In the second use case, does the sample of companies each year stays the same or it changes over time? If it changes over time, then one can’t say if the trend is changing because of the sample of earnings call transcript or some companies actually moving from not committing to climate goals to committing to them and vice versa. Also, it doesn’t list from which country these companies are located. The sharp increase since 2019 can be driven by just a change of policy in a few countries.
Q11: As earnings call is a quarterly reporting not yearly, how and why data was aggregated to a yearly frequency in Figure 1?
Q12: Why the guidelines listed in the appendix (lines 469-480) is not included in ChatGPT zero-shot prompt and rather a different prompt is used?
Q13: How does model performance ranking change if hyperparameters different from the one listed in Table C.2 are used?

**Reasons To Accept:**

This paper presents a refined dataset for a critical task and provides a comparative analysis of different models, including zero-shot ChatGPT. It also outlines two practical use cases, contributing to its acceptability.

**Reasons To Reject:**

Despite careful examination of the main paper and its appendix, I found that vital information about the data and its annotation is notably absent. This omission complicates the assessment of the paper's quality, hence my low score for soundness.

The paper's primary contribution lies in introducing/refining the dataset. Given the significance of this contribution, comprehensive details on the annotation process are necessary.

The paper needs a lot of writing work to address many questions I have listed in the section below.

**Reproducibility:**

3: Could reproduce the results with some difficulty. The settings of parameters are underspecified or subjectively determined; the training/evaluation data are not widely available.

**Reviewer Confidence:**

4: Quite sure. I tried to check the important points carefully. It's unlikely, though conceivable, that I missed something that should affect my ratings.

**Typos Grammar Style And Presentation Improvements:**

In general, the paper can be greatly improved by providing more detail on everything. Below are some specific comments:

C1: The abstract says corporates and nations while the “Data” section says companies, countries, regions, and cities.

C2: It is more conventional to write 1,500 instead of 1.500 (line #165).

C3: It will be better to include the ChatGPT prompt in the main paper instead of the Appendix. Even in the 4-page limit Limitations can be moved to the 5th page and the prompt could be included.

C5: More details can be added to the caption of Table 3 to improve the readability of the work.

C6: Data description for two use cases is missing in the main paper.

C7: Table H.4 can use better legends.

---

> ### Author Rebuttal · Authors · 2023-08-28
>
> Hello dear Reviewer,
>
> We are very grateful for the detailed feedback you provided. Since the feedback was very constructive, we tried to address the issues and the corresponding answers and analyses would also be part of the final paper. I will give a brief version of the answers.
>
> Answer to the questions:
> - Q1: The dataset contains 273 claims by cities, 1396 claims by companies, 205 claims by countries, and 159 claims by regions.
> - Q2: The descriptive statistics barely change after the preprocessing. We only exclude 97 data points (from 3614 to 3517) in the preprocessing. An overview of the descriptive statistics before and after preprocessing is added to the paper.
> - Q3, 4, 6: We not only collaborated with the Net Zero Tracker but one of the authors is also responsible for the data collection at the Net Zero Tracker. The data contained a collection of claims and the authors added an additional layer of labeling by checking each sample and the label. Thus, we can also create an inter-labeler agreement (Cohen's Kappa of around 93%). Upon disagreement, the data point was discussed with the team. Hence, at least two persons labeled a data point.
> - Q5: The annotators were employed by the Net Zero Tracker which is a combined initiative of Oxford Net Zero, the New Climate Institute, Data Driven Envirolab, and the Energy & Climate Intelligence Unit. Thus, the annotation steps are done by either the authors or persons with domain know-how.
> - Q7: For the additional human-in-the-loop quality check, we also used the five-fold cross-validation with ClimateBERT. We save the misclassifications of each fold and then an author decides whether the model is right or wrong. Thus, the model's decision boundary is always slightly altered and we reapt this process three times. This way, we assess samples where the model itself has trouble identifying the right label. Thus, we refine the model with the hard samples. This technique leans on Data Boosting which won an Andrew Ng challenge for deep vision systems.
> - Q8: The net-zero and reduction claim analysis in the earning calls is a use case, rather than an evaluation. Thus, the models are applied rather than tested with it and no accuracy is given. However, the patterns coincide with the general interest in sustainable investing since 2019 (see https://www.morningstar.com/sustainable-investing/2022-us-sustainable-funds-landscape-5-charts)
> - Q9: This was a fault, thank you.
> - Q10: We used the earning calls of all publicly listed US companies from 2003-2022. While I agree that the companies might change and there are likely some key events that influence the analysis (see link above), this use case mainly serves to show the large-scale analysis potentials and is not yet a full analysis for itself. It should give practitioners an idea of how the models could be used. Further descriptive statistics about for instance the number of events, companies, and analyzed sentences are added in the final paper version.
> - Q11: We build a yearly average of all quarters in a year to form a year index.
> - Q12: Initially, we wanted to create a simple prompt. I repeated the evaluation with the labeling guidelines in the prompt and achieved slightly better results. However, the overall picture of the BERT models outperforming ChatGPT stays consistent.
> - Q13: We've repeated the evaluation with a hyperparameter grid with the learning rates [3e-5, 5e-5, 7e-5], batch size of [16, 32], and epochs of [5, 10]. The results remained the same for all configurations with ClimateBERT outperforming the other models.
>
> Irrespective of the outcome of this review, we want to thank you for the good feedback. The underlying task is a very important real-world problem where advances are necessarily needed. Thus, we thank you for improving the work.

---

### Official Review · Reviewer_8w4H · 2023-08-01

**Soundness:** 4

**Excitement:**

4: Strong: This paper deepens the understanding of some phenomenon or lowers the barriers to an existing research direction.

**Missing References:**

I recommend citing and discussing some net-zero or greenwashing related papers.
Below is an example of papers I know of, but there must be others.

- An Integrated Framework to Assess Greenwashing. Sustainability. 2022.
- The meaning of net zero and how to get it right. Nature Climate Change. 2022.

**Paper Topic And Main Contributions:**

The major contribution of this paper is the proposed dataset.

Net-zero ("net zero" GHG emissions)  claims by firms and countries have been on the rise recently, but are sometimes pointed to as greenwashing.
This paper proposes a task and dataset for detecting net-zero claims from text.
The authors provided a 3.5K text annotated corpus including three labels: NetZero, Reduction, and NoTarget.
The fine-tuned ClimateBERT outperformed baselines.
The authors also experimented the reduction target year extraction by using QA approach.
Finally, a case study on the earning call transcripts analysis suggests that the net-zero target discussion has experienced a sharp increase since 2019.

**Questions For The Authors:**

- Any thoughts about A1 above?
- Any thoughts about A2 above?
- What is the nature of "expert annotator"?
- How did the authors decide the number of NoTarget samples.
- The evaluation results show extremely high classification performance of the model. This suggests that the task is extremely easy. Is it possible that the model is making decisions due to specific biases in the dataset?

**Reasons To Accept:**

- This paper proposes a novel task and dataset on a net-zero perspective of climate change. The dataset is valuable to researchers because this field is nascent and lacks datasets and task definitions. The authors will release the dataset and models, which will be useful for researchers in this domain.
- They appear to be collaborating with domain experts to create the dataset

**Reasons To Reject:**

The major drawback is that the paper lacks detail in the discussion and dataset.
It is especially important to discuss motives because this paper takes on a new challenge.
See below comments.

### High-level comment

A1. Lack of discussion for the motivation

The authors begin their motivation with a reference to greenwashing. Indeed, claims of net zero have sometimes been accused of greenwashing, so I agree with the importance of this motivation.
While the study seems to focus on extracting net-zero targets, there is no insight of what can be done to suppress greenwashing as a result.

There are various typologies of greenwashing, e.g., net zero may be dependent on carbon offsetting, there may be no clear intermediate target, it may not follow IPCC guidance, etc.

It is not clear what typologies of greenwashing the authors have in mind or whether they have extracted the necessary information to do so. It might be good to see these discussions in a related work, introduction, or discussion section.


A2. Lack of discussion for the task design

Similar to above, I would like to know the motivation behind the task definition.
The net-zero claim is a complex matter in practice. For example,
- Is the scope of the net-zero claim related to the company or for a specific project? Is it for Scope 1, 2, or 3 emissions?
- The difference between intermediary and long-term goals.
- Differences in units and calculation methods

The Appendix presents some of the guidelines, but they are oversimplified and lack an explanation of why the task was defined in the way.


### Low-level comment

B1. Detailed description of the dataset is missing.

- The label distribution for both train and test data should be described in Table 1.
- Please specify the nature of "expert annotator"
- The design of the NoTarget label is questionable. These samples may be different from the domains of the NetZero and Reduction labels. Thus, they may overestimate the classification performance of the model that can use such "bias". In fact, the models are extremely accurate. Furthermore, the reason for selecting the number of NoTarget labels is ambiguous. For example, there should be far more NoTarget labels in a sustainability report or other data source.
- The sample text of the dataset is not presented in the paper, making it difficult for readers to understand.
- It is not clear whether the annotation is in a paragraph or a sentence -level.

**Reproducibility:**

4: Could mostly reproduce the results, but there may be some variation because of sample variance or minor variations in their interpretation of the protocol or method.

**Reviewer Confidence:**

5: Positive that my evaluation is correct. I read the paper very carefully and I am very familiar with related work.

**Typos Grammar Style And Presentation Improvements:**

- By removing the bold text ("Motivation", "Contribution", "Results", and "Implications") in Introduction, more paper space can be used?
- Table 1 should include train and test statistics.

---

> ### Author Rebuttal · Authors · 2023-08-28
>
> Hello dear Reviewer,
>
> thank you very much for the detailed and comprehensive feedback. We find the points very good and will answer each of them briefly. All the background and details mentioned in the answers would also be included in the final paper version.
>
> Regarding A1 & A2 / Lack of discussion for the motivation & Lack of discussion for the task design:
> Indeed, the types of greenwashing are underspecified in the paper. We don't view ClimateBERT-NetZero yet as a tool for identifying greenwashing (of any type) but as a very important first step. This tool can help humans to fully or partially automate the net zero and reduction claim assessment. Thus, we aim to shed light on the fuzzy communication side. In the second step, this information has to be compared with real-world data (e.g., realized emissions) to identify greenwashing enterprises. With this paper and the publication of models and data, we want to extend the capabilities of a broader research and practice community. Hence, we also included the use cases of ambition assessment and large-scale analysis.
>
> Regarding Dataset issues (B1 and additional question):
> - To address the opacity of the label distribution, we added a Table with descriptive statistics of each validation set in the five-fol cross-validation. We used a stratified split. Therefore, the label distribution of the validation sets follows the general distribution (more or less 305 non-targets and 200 reduction and net zero targets).
> - Expert annotators are employed by the Net Zero Tracker which is a combined initiative of Oxford Net Zero, the New Climate Institute, Data Driven Envirolab, and the Energy & Climate Intelligence Unit. Thus, the annotation steps are done by either the authors or persons with domain know-how.
> - The number of No Target label sentences was set to balance the reduction and net zero claims. More specifically, it is mainly constructed with other climate-related sentences. The reason for this is that we propose to use the detection of climate-related sentences as introduced in the ClimateBERT paper as a preliminary step before finding net zero and reduction targets. So, this makes the actual real-world problem less imbalanced. However, also looking at the results, besides the accuracy, the F1 score is also very high indicating that all labels are predicted well. Actually, we also added the confusion matrix to the paper where we can see that the Non-Target is predicted with 99% accuracy giving evidence that the approach works particularly well on this important label.
> - We included sample texts in the paper to show what the labels look like. Furthermore, the classification problem is rather short paragraphs that include sentences.
>
> With the feedback of Reviewer 3, we addressed further data issues to make the contribution more solid. I refrain from restating them here but it complements your feedback and thus might be of interest to read for you.
>
> Irrespective of the outcome, I want to thank you again for providing feedback. As mentioned, the problem we're trying to solve is nothing abstract but rather a pressing real-world task. This is why we are happy that you made meaningful comments to improve this work.

---

### Official Review · Reviewer_Qtwv · 2023-08-01

**Typos Grammar Style And Presentation Improvements:** line 165
**Soundness:** 3

**Excitement:**

2: Mediocre: This paper makes marginal contributions (vs non-contemporaneous work), so I would rather not see it in the conference.

**Missing References:**

N/A

**Paper Topic And Main Contributions:**

In order to detect and assess the net zero and reduction targets, this paper introduces a classification model named ClimateBERT-NetZero, with ClimateBERT as the base model. This paper also introduces a dataset containing 3.5k text samples labeled by Net-Zero/Reduction/No Target as the dataset ClimateBERT finetuned on. Empirical results of this model applied on two subsequent tasks show the evidence of the effectiveness of the proposal of this paper.

**Questions For The Authors:**

[Question A]: To my understanding, ClimateBERT-NetZero is a classification model right? How do you apply it to the task of ambition prediction in a Q&A fashion?

**Reasons To Accept:**

1. The motivation of this paper seems clear and sound.
2. The empirical results also give evidence of the effectiveness of the proposal of this paper.

**Reasons To Reject:**

1. The data creation process lacks details. This paper put more details of the training data creation in the appendix, which is fine. But it never mentions anything about quality control (such as IAA), how samples in the dataset looks like.
2. This work lacks novelty. In my opinion, it just looks like an off-the-shelf BERT-based language model tuned on domain-specific data without any modification to make the model better fit this certain domain or task.

**Reproducibility:**

4: Could mostly reproduce the results, but there may be some variation because of sample variance or minor variations in their interpretation of the protocol or method.

**Reviewer Confidence:**

3: Pretty sure, but there's a chance I missed something. Although I have a good feel for this area in general, I did not carefully check the paper's details, e.g., the math, experimental design, or novelty.

---

> ### Author Rebuttal · Authors · 2023-08-28
>
> Hello dear Reviewer,
>
> thank you for your feedback. Indeed, ClimateBERT-NetZero "just" serves the task of identifying reduction and net zero claims. However, we argue that this task and dataset represent an important contribution because they address an important real-world problem with much-needed nice language, i.e. climate change disclosure language. In an abundance of specific datasets in the realm of climate change, corresponding challenges cannot be solved. Specifically, adaptation and transition disclosures (i.e., net zero and reduction ambitions) will play an essential role in the real-world economy and in fighting climate change. Thus, we build on existing work in this domain (e.g. ClimateBERT) to create the best possible domain-specific solution.
>
> In acknowledgment of your comments, we created new metrics and information for the data creation process that would be included in the final version of the paper. I will briefly outline three central components that address your points about data quality and transparency. We add a more detailed view of the data quality and make the data creation process more transparent by including details on the dataset creation. One label was assigned by the Net Zero Tracker and controlled by the author team to reach an inter-labeler agreement. This agreement is very high with a Cohen's Kappa of 93%. When disagreements arose, they were discussed with the team. Second, more details on the additional human-in-the-loop control process could mitigate concerns about data quality problems. One central element that is further described is that we used the five-fold cross-validation with ClimateBERT and saved all misclassifications of each fold. Then, the author team discussed them and thereby refined the dataset. This way, the dataset ran through multiple human and machine checks. Third, we included additional multiple statistics about text length before and after preprocessing, the label distributions during training and validation as well as samples of the dataset in the paper to address your concerns about the opacity of how the dataset looks.
>
> Regarding your question (Question A): we don't use ClimateBERT-NetZero for the QA approach. As described in the paper, we use Roberta-base-squad2 and create the evaluation dataset for the questions.
>
> We hope these comments clarify some concerns about the paper.

---

### Meta-Review · Area_Chair_krQQ · 2023-09-16

**Recommendation:** 4

**Metareview:**

This paper proposes ClimateBERT-NetZero, a natural language classifier for detecting corporate and national net zero and emission reduction targets from the text. The key contribution is a new expert-annotated dataset of 3.5K text samples categorized as NetZero, Reduction, or NoTarget claims. Authors have provided experiments which demonstrate ClimateBERT outperforms baselines when fine-tuned on this data. Additionally, they provided two use cases highlighting the potential to extract and analyze climate commitments at scale.

The motivation to assess ambiguous sustainability claims is timely and important. Reviewers praise the novelty of the task and release of models/data but raise issues about dataset details and reproducibility. In particular, Reviewer 2 gave very comprehensive feedback on framing the work and recycled interventions. In response, the authors sufficiently clarify the annotation process, data statistics, and human-in-the-loop mechanisms. Additional analyses also confirm model performance on real-world transcripts.

Personally, I think that the task is technically trivial, however, the significance and impact of the studies are meritorious. Moreover, I agree with the reviewers that the paper is fundamentally sound even though I am of the opinion that the paper would benefit from more discussion situating the work amidst greenwashing typologies and nuances of claim verification as also implied by one of the reviewers. Also, the current heavy focus on introducing the dataset leaves the overall contributions feeling somewhat incremental. Providing sample texts and better highlighting incremental contributions over prior climate NLP would strengthen the manuscript.  Finally, more robust framing, analysis, and comparisons to prior climate NLP could strengthen the clarity and impact.

---

### Decision · Program_Chairs · 2023-10-07

**Decision:**

Accept-Main

**Comment:**

This paper proposes ClimateBERT-NetZero, a natural language classifier for detecting corporate and national net zero and emission reduction targets from the text. The key contribution is a new expert-annotated dataset of 3.5K text samples categorized as NetZero, Reduction, or NoTarget claims. Authors have provided experiments which demonstrate ClimateBERT outperforms baselines when fine-tuned on this data. Additionally, they provided two use cases highlighting the potential to extract and analyze climate commitments at scale.

The motivation to assess ambiguous sustainability claims is timely and important. Reviewers praise the novelty of the task and release of models/data but raise issues about dataset details and reproducibility. In particular, Reviewer 2 gave very comprehensive feedback on framing the work and recycled interventions. In response, the authors sufficiently clarify the annotation process, data statistics, and human-in-the-loop mechanisms. Additional analyses also confirm model performance on real-world transcripts.

Personally, I think that the task is technically trivial, however, the significance and impact of the studies are meritorious. Moreover, I agree with the reviewers that the paper is fundamentally sound even though I am of the opinion that the paper would benefit from more discussion situating the work amidst greenwashing typologies and nuances of claim verification as also implied by one of the reviewers. Also, the current heavy focus on introducing the dataset leaves the overall contributions feeling somewhat incremental. Providing sample texts and better highlighting incremental contributions over prior climate NLP would strengthen the manuscript.  Finally, more robust framing, analysis, and comparisons to prior climate NLP could strengthen the clarity and impact.